# Prescribing Pattern and Safety Profile of Biological Agents for Psoriasis in Real-World Practice: A Four-Year Calabrian Pharmacovigilance Analysis

**DOI:** 10.3390/pharmaceutics16101329

**Published:** 2024-10-14

**Authors:** Caterina De Sarro, Francesca Bosco, Agnese Gagliardi, Lorenza Guarnieri, Stefano Ruga, Antonio Fabiano, Laura Costantino, Antonio Leo, Caterina Palleria, Chiara Verduci, Vincenzo Rania, Michael Ashour, Luca Gallelli, Rita Citraro, Giovambattista De Sarro

**Affiliations:** 1Department of Health Sciences, Magna Graecia University of Catanzaro, 88100 Catanzaro, Italy; catedesarro@gmail.com (C.D.S.); francesca.bosco@unicz.it (F.B.); gagliardi@unicz.it (A.G.); lorenza.guarnieri@unicz.it (L.G.); rugast1@gmail.com (S.R.); a.fabiano@unicz.it (A.F.); laura.costantino@studenti.unicz.it (L.C.); aleo@unicz.it (A.L.); palleria@unicz.it (C.P.); verducichiara@icloud.com (C.V.); raniavincenzo1@gmail.com (V.R.); a.michael@unicz.it (M.A.); gallelli@unicz.it (L.G.); desarro@unicz.it (G.D.S.); 2System and Applied Pharmacology@University Magna Grecia, (FAS@UMG) Research Center, Science of Health Department, School of Medicine, Magna Graecia University of Catanzaro, 88100 Catanzaro, Italy

**Keywords:** psoriasis (PS) treatment, biologicals, pharmacovigilance, adverse events (AEs), safety

## Abstract

Background: The treatment of psoriasis has made considerable progress with biologicals, including tumor necrosis factor inhibitors, and recently, monoclonal antibodies inhibiting directly interleukin (IL) 17, IL-23, or both IL-12/23. Newer biologicals are directed to the interleukin pathway and appear to improve complete or near-complete clearance. The newer biologicals have also been shown to have an excellent safety profile. However, despite experience with patients having confirmed the results obtained in clinical trials, there are still few data on using the newer biologicals. Methods: The present active study aimed to prospectively evaluate safety profiles and persistence of some biologicals in a multicenter pharmacovigilance study, that enrolled 733 patients treated with a biologic drug in five Calabrian hospital units. Informative and treatment persistence evaluations with predictors for suspension and occurrence of adverse events (AEs) were executed. In particular, reasons for treatment discontinuation in our program take account of primary/secondary failure or development of an AE. Results: AEs occurred in 187/733 patients and serious AEs (SAEs) were identified in 5/733 patients. An number of 182/733 patients showed a primary/secondary inefficacy. The AEs and SAEs were described with adalimumab, infliximab, and etanercept but not with abatacept, brodalumab, tildrakizumab, golinumab, ixekizumab, guselkumab, risankizumab, secukinumab, and ustekinumab. Conclusions: Our analysis, although limited by a small sample size and a short-term follow-up period, offers suitable data on commonly used biological agents and their safety, interruption rate, and the attendance of SAEs. Real-world studies should be carried out to evaluate other safety interests.

## 1. Introduction

Psoriasis (PS) is a common chronic inflammatory skin disease that affects about 3% of the worldwide population [1] and is linked with physical and psychological burdens worsening patients’ value of life [2]. Although the aetiology of PS is still unclear, it is now considered a complex disease characterised by the interaction between susceptibility hereditary alleles and environmental risk factors, understood as disease triggers (stress, obesity, bacterial and viral infections, trauma, smoking, and alcohol consumption) [3,4]. Both PS and psoriatic arthritis (PsA) are capable of compromising patients’ quality of life (QoL) and with extremely complex clinical pictures. The most common form of PS is plaque PS, which generally manifests as skin inflammatory plaques, the onset of which correlates with an excessive production of inflammatory cytokines. More precisely, in this phase, myeloid dendritic cells, through the production of IL-12 and IL-23, induce the activation of T-helper cells, which are predisposed to the production of IL-17, tumor necrosis factor-alpha (TNF-α), interferon-gamma (IFN-γ), and IL-22. This leads to premature maturation of dendritic cells and keratinocytes, as well as to leucocyte infiltration and vasodilatation, resulting in epidermal hyperproliferation and plaque formation [5,6]. Therefore, PS can be characterised by diverse manifestations and various chronic diseases. These have a higher occurrence in people with PS such as PsA (10–30% incidence), cardiovascular diseases, depression, and Crohn’s disorder, increasing morbidity and mortality of patients [7,8]. The complexity of these manifestations can produce health issues, including physical and mental disability [9,10]. However, in agreement with evidence-based PS guidelines [11,12], topical treatments, phototherapy, and biological and conventional drugs are currently used to treat PS and PsA [13].

The European Guidelines for PS [14] suggest starting with the use of the conventional synthetic Disease-Modifying Antirheumatic Drug (csDMARDs) methotrexate, cyclosporine A, retinoids, and derivates of fumaric acid and in cases of inadequate response, contraindication, or intolerance to at least one csDMARD, to then administer treatment with biologic DMARDs (bDMARDs).

In the last twenty years, clinical therapy has been modernised after discovering and marketing various new biological agents, resulting in a marked advance in managing recalcitrant patients [15]. These drugs may selectively affect the action of specific mediators implicated in the start and maintenance of inflammatory activities. At present, among biological drugs, four TNF-α antagonists (adalimumab-ADA; etanercept-ETN; and golimumab-GOL, for subcutaneous administration; infliximab-IFX for intravenous infusion), two IL-17A inhibitors (secukinumab-SEC, ixekizumab-IXE), one IL-17 inhibitor (brodalumab-BRO), one IL-12/23 inhibitor (ustekinumab-UST), three IL-23 inhibitors, (guselkumab-GUS, risankizumab-RZE, tildrakizumab-TIL), one IL-6 receptor antagonist (tocilizumab-TCZ), and one T-cell co-stimulation inhibitor (abatacept-ABT) are used for both PS and PsA. In detail, all TNF-α inhibitors are approved for either PS and PsA and, in particular, ADA is also approved for Hidradenitis Suppurativa (HS); SEC, ABT, TCZ, GUS, UST, BRO, IXE, RZE, and TIL are also approved for PsA and PS. However, side effects such as infections, hypersensitivity reactions, and some solid tumor types have been associated with this type of drug. Therefore, careful post-marketing monitoring and patient selection are crucial. This active observational and prospective post-marketing study aims for both physician and clinical pharmacologist to monitor and analyze suspected AEs following the above-reported biological agent’s therapy with a detailed description in five dermatologic units of the active Calabria Biologics Pharmacovigilance Program (CBPP) [16].

## 2. Materials and Methods

Patients who satisfied the inclusion criteria, treated with a biologic drug for 4 years (2017–2021), were enrolled from hospitals in Calabria in 5 dermatology units. For each patient, demographic and clinical data, current therapies, any discontinuations, failures, switches, and AEs recorded in the two years preceding the following procedure were collected. A clinical pharmacology specialist with knowledge of pharmacovigilance was paired with the dermatology physician to build the database following Good Clinical Practice guidelines. At the same time, a patient-encrypted code was used to preserve the privacy of patients in agreement with the Declaration of Helsinki (1964) and its subsequent revisions. The study protocol was accepted by the local Ethics Committee (Comitato Etico Regionale Calabria, Italy), protocol number 278/2015.

### 2.1. Study Design and Data Collection

The CBPP is a multicenter pharmacovigilance study focused on increasing the constant observation of the safety of therapy with biological agents in real-world practice. Additionally, as described before, the study offers an initial education training phase to physicians on pharmacovigilance data and AE correct documentation (15). Records have been achieved for four years of the CBPP for the evaluation of the security and pertinence of biological treatment in dermatologic units. All following patients undergoing treatment with one biological agent at 5 tertiary dermatology outpatient hospital centers (Azienda Ospedaliera Universitaria “Mater Domini”, Catanzaro, Italy, Azienda Ospedaliera “Pugliese-Ciaccio”, Catanzaro, Italy; Azienda Ospedaliera Cosenza, Cosenza, Italy; Grande Ospedale Metropolitano “Bianchi-Melacrino-Morelli”, Reggio Calabria, Italy; Azienda Ospedaliera Provinciale Crotone, Crotone, Italy) between 1 January 2017 and 1 March 2021, have been selected for eligibility.

### 2.2. Inclusion Criteria

All 733 patients (571 naïve patients and 162 patients already treated with biological agents) enrolled in this study fulfilled the following inclusion criteria: age > 18 years; diagnosis of moderate to severe PS or PsA and treatment with a biological agent; informed consent was obtained from all enrolled patients at the time of admission. The index date was defined as the start of biological therapy during the study protocol for each patient naive or previously treated with a biological agent. A hybrid system for detecting side effects during the follow-up period from the index date was used: telephone calls from the pharmacologist, as previously reported, and routine specialist visits. The data collected from each associated patient included demographic and clinical data such as age, diagnosis, gender, recent or previous corticosteroid therapy, duration of disease, biologic or biosimilar agents, DMAR, additional drugs, probable discontinuation, or switch to another agent with a specific rationale, significant comorbidities, possible primary or secondary therapeutic failure, and occurrence of AEs. Despite this growing therapeutic armamentarium, not all patients respond to biological drugs, as patients who initially respond to therapy may lose response to treatment or even develop side effects over time. The motivation for therapy discontinuation (withdrawal with or without therapy switch) in our program involved primary/secondary failure (i.e., inability to respond in 16 weeks or no therapeutic effects to a new biological agent used or) or the development of AE. Approved clinical pharmacologists helped physicians to characterise primary or secondary therapeutic flops and probable AEs, through a correct evaluation of clinical reports. 

### 2.3. Data Collection

For every AE noted, the medical doctor loaded in the suspected AE related form of the Italian AIFA. AEs were classified following MedDRA (Medical Dictionary for Regulatory Activities) terminology by reporting the time of onset, recovery, gravity, and termination. In cases where AEs were defined as severe (example in life-threatening cases), switching from one biologic to another was required. Switching biologic is defined as the replacement of one biologic by another within the same class situation, a switch out of class is called a swap. Today, the switch also relates to any change from the originator biologic (“reference product”) to its biosimilar. Biosimilars are similar copies of complex biologic medicines, equally effective and safe but usually at lower prices.

### 2.4. Statistical Analysis

The sample size was built on the patient’s admission to each research site and not predesignated (i.e., randomised recruitment). An informative study was carried out to condense the basal and demographic properties of admitted patients at the index date. Continuous data are given as mean standard deviation (SD) or median (25–75 percentile) as applicable, whilst categorical records are articulated as numbers (percentage). A specific treatment was assigned for each patient. Medians were quantified with interquartile ranges (Q1–Q3) for permanent and pure variables, while percentages of occurrence were used to evaluate categorical variables. SPSS software (IBM Corp. SPSS Statistics, Armonk, NY, USA) version 27.0 was used for statistical analyses. The Kolmogorov–Smirnov test was used to identify normality. Continuous variables were examined with the Mann–Whitney U test, while for categorical variables, two-tailed Pearson’s chi-squared test was performed. The frequency of some ADRs was analyzed to calculate their severity, type, and switch/swap of the various drugs. The results were shown as hazard ratios (HR) with corresponding 95% confidence intervals (CI). Values with *p* < 0.05 were considered significant.

## 3. Results

### 3.1. General Characteristics of the Study Population

Table 1 reports the general characteristics of the CBPP patients of the present study. During the study period, 733 patients (303 females; male 430, mean age 54.7 ± 13.4) were started on treatment with biologic drugs for active PS (n = 353, 48.1%), PsA (n = 380, 51.9%), at five dermatology centers distributed across the Calabria Region (Southern Italy). The proportion of patients affected by plaque psoriasis (PP) was more frequent in younger patients than older ones, while PsA was more common in older adults. Comorbidities were more common in patients 65 years or older. Adalimumab (ADA) was the most commonly prescribed biologic drug at the index date (230; 31.3%), followed by etanercept (ETN) (219; 29.8%), ustekinumab (UST) (94; 12.8%), infliximab (IFX) (48; 6.5%), secukinumab (SEC) (65; 8.8%), golimumab (GOL) (32; 4.3%), ixekizumab (IXE) (18; 2.4%), guselkumab (GUS) (9; 1.2%), risankizumab (RZE) (7; 0.9%), brodalumab (BRO) (6; 0.8%), tildrakizumab (TIL) (3; 0.4%) and abatacept (ABT) (1; 0.1%). Data subdivided per drug are reported in Table 2 and Table 3.

Moreover, 255 patients (33.7%) received concomitant treatment with one or more immunomodulatory drugs, methotrexate (MTX) (96; 12.6%), cyclosporine A (CYA) (76; 10%), systemic and topical corticosteroids (CCS) (50; 6.6%), nonsteroidal anti-inflammatory drugs (NSAIDs) (2; 3.7%), or acitretin (ACI) (5.0; 6%). Characteristics of study cohort of patients affected by PS per biological and non-biological agents are detailed and reported in Table 3. The 77% of patients were naive to biologic therapy at the index date. The remainder underwent a switch/swap with a previous biologic drug. In addition, Table 3 clearly describes the number and the percentage of patients who maintained therapy or had a switch or a swap.

### 3.2. Switches Related to Inefficacy and AEs

A clear distinction must be made between AEs and ineffectiveness. A specific analysis was performed for the switches or swaps with the related reasons, in particular, 182 switched/swapped from one or more former biological agents (number of former biological agents range 1–3), of which 150 for ineffectiveness and 29 AEs and 3 SAEs as reported in Table 4.

### 3.3. Descriptions of AEs

During the study time, we observed 187 (25.5%) AEs and 5 (0.66%) SAEs, for a total of 192 (25.7%) adverse events. In our cohort for each drug, AEs/SAEs have been reported with GOL (11/32; 34.3%), BRO (2/6; 33.3%), TIL (1/3; 33.3%), ADA (69/230; 29.5%), IXE (5/18; 27.7%), ETN (54/219; 24.6%), IFX (11/48; 22.9%), SEC (12/65; 18.4%), RZE (1/7; 14.2%), UST (11/94; 11.7%); no AEs have been occurred with GUS (0/9), ABT (0/1), and TCZ (0/1). The most common adverse events were asthenia, injection site reactions, skin disorders, and gastrointestinal disorders, all the AEs discerned were awaited. An extensive list of AEs and SAEs categorised according to the MedDRA dictionary is presented in Table 5.

During the study period, we noted five patients experiencing SAEs: etanercept caused three cases of SAEs (1.3%), specifically one case of the new-onset, lupus-like syndrome, one case of benign respiratory tract neoplasm, and one of hemorrhagic cystitis; infliximab caused one case of SAE (0.4%). A case leading to hospitalization following severe splenomegaly occurred after adalimumab (0.4%) administration.

## 4. Discussion and Conclusions

The present active post-marketing report is the first prospective study aimed at investigating both the persistence and safety profiles of biological agents for 4 years in the Calabria region. Patients affected by PS enrolled in the present study occurred with greater frequency in men (58.6%) which agreed with findings in other reports (16). The development and marketing of several bDMARDs have radically changed the therapeutic approach to PS and PsA. These drugs have proven their efficacy and safety profile in the long term, owing to the absence of cumulative organ toxicity. Thus, they represent an excellent therapeutic tool for the long-term continuous therapeutic management of psoriatic patients [17,18]. Data from the literature suggest that most patients are likely to maintain a good safety and efficacy profile over the years [19,20]. The index data, demonstrated that ADA and ETN were the most prescribed first- and second-line drug treatments due to the availability of less costly biosimilars, followed by UST, SEC, IFX, and GOL. Biological agents have usually various improvements associated with traditional therapies together with no data of growing toxicity or scientific-related drug–drug relations, reputable long-term effectiveness, and reasonable utilization in hepatic or renal impaired patients [21]. Data obtained from the open-label extension phase of clinical trials and data obtained from real life often do not overlap. This discrepancy depends mainly on the fact that the PASI75 endpoint does not correlate perfectly with patients’ perceptions of treatment efficacy [22].

Nevertheless, sporadic and changeable AEs are arduous to identify in pre-marketing clinical trials because of inclusion conditions and modest sample dimensions. Moreover, the AEs associated with biological agents may not be related to their restricted activity, but due to the eliciting of undesirable immune reactions, through anti-drug antibody creation [23]. Various unprompted reporting treatments carried out during phase IV analysis might offer interesting data on tolerability, long-term usefulness, and demonstrating thinkable judges of ineffectiveness, AEs, and/or SAEs appearance [24].

However, the occurrence of AEs with the use of biological agents, in addition to other drugs, is far from ideal, with extensive under-recording, and currently, the spontaneous recording of AEs has miscalculated the actual number of AEs [25].

### 4.1. Importance of Active and Prospective Pharmacovigilance Studies

Effective pharmacovigilance events might be more valuable for finding and describing severe and unpredicted AEs [26,27,28]. Therefore, to detect early known and still unknown AEs in clinical practice, active pharmacovigilance programs have gained rising importance. Spontaneous reporting of suspected AEs represents the cornerstone of pharmacovigilance, but it is limited by a high under-reporting rate, as has been demonstrated [29]. In this real-world study, we analyzed data to improve the quality and quantity of AE reporting associated with biologicals in PS and PsA. The first important observation was that nearly the same proportion of patients suffering AEs have been described with each biological agent, except for GOL. The low number of patients receiving GOL in our cohort merits further consideration after an increase in the number of patients. Our second findings show only 0.6% of SAEs, require hospitalization, or with clinically relevant conditions. The most popular AEs in the present report belong to common ailments, in particular asthenia and injection site reaction (mild to moderate), gastrointestinal ailments, specifically infectious diarrhoea, and nervous system disorders that start with headache, followed by surveys. anomalies. In patients treated with TNF-α inhibitors, an increased rate of infection was found, the most commonly reported ones were upper respiratory tract infections (pharyngitis and sinusitis) [30], and an increased risk of opportunistic infections. Specifically, in patients not on coexisting corticosteroid therapy, we observed only 11 cases of non-serious infections, four herpes simplex diseases and three candidiasis, two cases of pneumonia (one of which was severe), and four cases of rhino-pharyngitis. In addition, there was one case of pharyngitis in a patient taking TNF-α inhibitors. Rare opportunistic infections or reactivation of tuberculosis appeared in no cases. Five cases of SAE including a neoplasm in the lung tract and a typical lupus-like disease in patients taking TNF-α inhibitors have been recorded [28]. However, Kimball and coworkers [29] reported that the likelihood of neoplasia was more likely but not associated with biologic therapy in PS patients.

### 4.2. Phenomenon of Switch/Swap in Our Real-World Study

In the context of medical practice, there are patients for whom biologic drugs show limited efficacy from the beginning or lose efficacy after an initial prolonged positive response. In addition, efficacy may decline when the original drug is resumed after a period of discontinuation of therapy. In such situations, a decision may be made to discontinue the initial biologic or switch to another biologic drug. In our study, we observed that 182 patients had switched from one or more previous biologic drugs (1 to 3). The main reasons for switching off the first biologic were secondary loss of efficacy and primary inefficacy, which together caused more than 82% of the changes in therapies. AEs are attributed to only 17.2% of therapy changes.

### 4.3. Conclusions

The evolution of PS treatment over the last five decades is a testament to the progress made so far in both understanding and management of this disease. A collaborative approach among physicians, patients, academics, and the pharmaceutical industry has been instrumental in enabling this progress. Although a cure for PS is unlikely anytime soon, complete elimination of the disease with the latest biologic therapies is now a realistic goal for some. An approach to disease management that embraces the P4 medicine principles of prediction, prevention, individualised therapy, and patient participation is a logical extension of our realization that PS is a “systemic disease” with important physical and psychosocial consequences. Although PS treatment has advanced considerably, there is still an unmet need for more effective therapies. The development of multiple potent biologic agents and small molecule inhibitors has expanded treatment regimens and transformed patient outcomes for PS. Biologicals have been highly effective in make better PS scores and improving and controlling psoriatic arthritis symptoms, helping patients dramatically increase their quality of life.

Since PS and PsA are chronic diseases and treatment with biologicals is usually performed for a long period (several years), efficacy and safety must be evaluated in the long term. In the present study, the AEs reported in the period 2017–2021 were overall comparable to those reported in other studies in the literature [30]. Our results show that biologicals for the treatment of PS and PsA are generally effective and well tolerated; in fact, they are in agreement with other studies already reported in the literature. Here, not many serious AEs were collected [31,32]. In a multicenter observational study conducted in 2016 on patients similar to ours, regarding the safety and efficacy of ADA, the authors observed an incidence of SAEs equal to 3.3% for ADA which in our study is the most prescribed drug [33]. However, the data we have collected allow us to be more optimistic in terms of safety as we have observed a lower number of SAEs for this biologic (0.4%) than previously reported.

The limitations of our findings must be considered: the most evident is the small trial amount per agent that reduces the probability of finding variations among therapies regarding switch/swap or analysing all the possible causes that might affect AE insurgence. Our analysis supports useful records on generally used biological compounds and their admissibility, stoppage amount, and the occurrence of SAEs. Additional studies must involve a plan for long-term follow-up [34]. Our post-marketing surveillance actions play an important role in significantly educating the identification and reporting of AEs and SAEs in a real-life context, providing important data on the safety of numerous treatments.

## Figures and Tables

**Table 1 pharmaceutics-16-01329-t001:** General characteristics of the patients of the present study.

Age of Patients	<35	35–49	50–64	>65	Naive Patients	Patients PS	Patients PsA
Sex					571	353	380
Male	31	130	169	100	341	211	219
Female	13	78	134	78	230	142	161
Pt. naïve	39	184	273	75			
Comorbidities	9	46	74	126			

Abbreviations: PS, psoriasis; PsA, psoriatic arthritis.

**Table 2 pharmaceutics-16-01329-t002:** Characteristics of the study cohort.

Overall Patients (n = 733)	
**Age, years**	
Mean (±SD)	54.7 ± 13.4
Range (minimum–maximum)	(min 19–max 91)
Median age (IQ range)	56
**Sex**	
Female, n (%)	303 (41.3)
Male, n (%)	430 (58.6)
**Mean age first biologic therapy, years (±SD)**	
	51.7 ± 13.7
**Nai** **¯ve, n (%)**	
	571 (77.9)
**Diagnosis**	
Psoriatic arthritis, n (%)	380 (51.8)
Plaque psoriasis, n (9%)	353 (48.1)
**Biologic drugs prescribed**	
IFX, n (%)	48 (6.5)
ADA, n (%)	230 (31.3)
GOL, n (%)	32 (4.3)
ETN, n (%)	219 (29.8)
UST, n (%)	94 (12.8)
SEC, n (%)	65 (8.8)
ABT, n (%)	1 (0.1)
TCZ, n (%)	1 (0.1)
BRO, n (%)	6 (0.8)
GUS, n (%)	9 (1.2)
IXE, n (%)	18 (2.4)
RZE, n (%)	7 (0.9)
TIL, n (%)	3 (0.4)
**Biosimilars, n (%)**	23 (5.2)
ADA, n (%)	11 (2.4)
ETN, n (%)	12 (2.6)
**Concurrent treatments**	
MTX, n (%)	96 (12.6)
CyA, n (%)	76 (10)
CCS, n (%)	50 (6.6)
NSAIDs, n (%)	28 (3.7)
ACI, n (%)	5 (0.6)
**Switched, n (%)**	182 (24.8)
Switch, n (%)	97 (53.3)
Swap, n (%)	85 (53.1)
**Adverse events**	
AEs, n (%)	186 (25.3)
SAEs, n (%)	5 (0.6)
Inefficacy, n (%)	150 (20.4)

Abbreviations: IFX, infliximab; ADA, adalimumab; GOL, golimumab; UST, ustekinumab; ETN, etanercept; SEC, secukinumab; ABT, abatacept; TCZ, tocilizumab; BRO, brodalumab; GUS, guselkumab; IXE, ixekizumab; RZE, Risankizumab; TIL, tildrakizumab; yrs years. MTX, methotrexate; CyA, cyclosporine A; CCS, corticosteroids; NSAIDs, non-steroidal anti-inflammatory drugs; ACI, acitretin; n, number.

**Table 3 pharmaceutics-16-01329-t003:** Characteristics of study cohort of patients affected by psoriasis for bDMARDs and scDMARDs.

	IFX	ADA	GOL	UST	ETN	SEC	ABT	TCZ	BRO	GUS	IXE	RZE	TIL
	(n = 48)	(n = 230)	(n = 32)	(n = 94)	(n = 219)	(n = 65)	(n = 1)	(n = 1)	(n = 6)	(n = 9)	(n = 18)	(n = 7)	(n = 3)
**Age, years**													
**Mean (±SD)**	55.3 (±12.5)	55 (±13)	53.9 (±13.8)	54.1 (±14.4)	56.3 (±12.8)	55 (±12.6)	59	63	50 (±9)	5.9 (±16.9)	51.5 (±17.1)	51.4 (±13.7)	41 (±7.5)
**Range (minimum–maximum)**	28–83	28–83	21–78	24–84	20–84	30–84	-	-	43–67	22–77	23–91	26–67	33–48
**Median age (IQ range)**	55.7 (46.9–63)	55 (46–65)	56 (44.8–64)	56 (43.2–64)	57 (48.1–66)	56.7 (45–63)	-	-	46.5 (44.5–51.5)	49 (42–51.5)	54.5 (43.3–62.8)	52 (46.5–61)	42 (37.5–45)
**Sex**													
**Female, n (%)**	20 (41.6)	102 (44.4)	19 (59.3)	37 (39.3)	85 (38.8)	2.4 (36.9)	0	1	1 (16.6)	3 (33.3)	8 (44.4)	3 (42.8)	1 (33.3)
**Male, n (%)**	28 (58.3)	128 (55.5)	13 (40.6)	58 (61.7)	134 (61.1)	41 (63)	1	0	5 (83.3)	6 (66.6)	10 (55.5)	4 (57.1)	2 (66.3)
**Mean age first biologic therapy, years (±SD)**													
	50 (±13.4)	52 (±13.3)	51.8 (±13.9)	51.3 (±14.8)	51.8 (±13.1)	53.2 (±12.2)	56	53	50.2 (±9.6)	51.1 (±17.3)	50.8 (±17.6)	51.4 (±13.7)	41 (±7.5)
**Nai** **¯ve, n (%)**													
	31 (64.5)	189 (82.1)	22 (68.7)	79 (84)	147 (67.1)	53 (81.5)	1	1	3 (50)	2 (22.2)	13 (72.2)	4 (57.1)	3 (100)
**Diagnosis**													
**Psoriatic arthritis,** **n (%)**	25 (52)	125 (53.4)	31 (96.8)	33 (35.1)	133 (60.7)	23 (35.3)	1	1	1 (16.6)	2 (22.2)	5 (27.7)	0	0
**Plaque psoriasis,** **n (%)**	23 (48)	105 (44.8)	1 (3.1)	61 (64.8)	86 (39.2)	42 (64.6)	0	0	5 (83.3)	7 (77.7)	13 (72.2)	7 (100)	3 (100)
**Concurrent treatment**													
**MTX, n (%)**	16 (33.3)	32 (13.6)	13 (40.6)	2 (2.1)	21 (9.5)	11 (16.9)	0	0	0	0	2 (11.1)	0	0
**CyA, n (%)**	15 (31.2)	15 (6.4)	1 (3.1)	7 (7.4)	29 (13.2)	3 (4.6)	0	0	0	0	0	0	0
**CCS, n (%)**	0	15 (6.4)	6 (18.7)	1 (1)	10 (4.5)	12 (18.4)	0	1	0	2 (22.2)	2 (11.1)	0	0
**NSAIDs, n (%)**	0	12 (5.1)	4 (12.5)	0	5 (2.2)	4 (6.1)	0	0	0	0	2 (11.1)	1 (14.1)	0
**Acitetrin, n (%)**	0	1 (0.4)	0	1 (1)	3 (1.3)	0	0	0	0	0	0	0	0
**Switched, n (%)**													
	26 (54.1)	34 (14.5)	11 (34.3)	9 (9.5)	60 (27.3)	10 (15.3)	0	0	3 (50)	2 (22.2)	4 (22.2)	2 (28.5)	0
**Switch, n (%)**	15 (57.6)	18 (7.6)	12 (37.5)	2 (22.2)	43 (71.6)	2 (20)	0	0	2 (66.6)	0	2 (50)	1 (50)	0
**Swap, n (%)**	14 (53.8)	16 (6.8)	2 (6.2)	9 (100)	27 (45)	11 (100)	0	0	2 (66.6)	2 (100)	2 (50)	2 (100)	0
**Biosimilars, n (%)**													
**ADA, n (%)**	0	11 (4.7)	0	0	0	0	0	0	0	0	0	0	0
**ETN, n (%)**	0	0	0	0	12 (5.4)	0	0	0	0	0	0	0	0
**Adverse events**													
**AEs, n (%)**	10 (20.8)	68 (29.5)	11 (34.3)	20 (21.7)	51 (23.1)	12 (17.1)	0	0	2 (33.3)	1 (11.1)	5 (27.7)	1 (14.2)	1 (33.3)
**SAEs, n (%)**	1 (2)	1 (0.8)	0	0	3 (1.3)	0	0	0	0	0	0	0	0

Abbreviations: IFX, infliximab; ADA, adalimumab; GOL, golimumab; UST, ustekinumab; ETN, etanercept; SEC, secukinumab; ABT, abatacept; TCZ, tocilizumab; BRO, brodalumab; GUS, guselkumab; IXE, ixekizumab; RZE, Risankizumab; TIL, tildrakizumab; yrs years. MTX, methotrexate; CyA, cyclosporine A; CCS, corticosteroids; NSAIDs, non-steroidal anti-inflammatory drugs; ACI, acitretin.

**Table 4 pharmaceutics-16-01329-t004:** Switches related to inefficacy (switches related to AEs), (SAE).

Switch/Swap to											
Switch/Swap From	IFX	ADA	GOL	UST	ETN	SEC	BRO	GUS	IXE	RZE	TIL
IFX		6 (1)	-	-	8 (1)	-	-	-	-	-	-
ADA	6 (5)		3 (2)	3	23 (5)	1	-	2	1 (1)	1	-
GOL	5 (1)	3 (1)		-	-	1 (2)	-	-	1	-	-
UST	6	13 (1)	-		12 (1)	-	-	-	-	1	-
ETN	3 (1)	9 (7)	5 (1)	3 (1)		5 (2)	1				1
SEC	6	3 (1)	-	4	3		-	-	1 (1)	-	-
BRO	-	-	-	-	-	-		-	-	-	-
GUS	-	-	-	-	-	-	-		-	-	-
IXE	-	-	-	-	-	-	2	-		-	-
RZE	-	-	-	-	-	-	-	-	-		-
TIL	-	-	1	-	1	-	-	-	-	-	

Abbreviations: IFX, infliximab; ADA, adalimumab; GOL, golimumab; UST, ustekinumab; ETN, etanercept; SEC, secukinumab; IXE, ixekizumab; GUS, guselkumab; RZE, Risankizumab; BRO, brodalumab; TIL, tildrakizumab.

**Table 5 pharmaceutics-16-01329-t005:** MedDRA-compliant description adverse events (AEs).

MedDRA-Compliant Description AEs	IFX	ETN	ADA	GOL	UST	SEC	BRO	GUS	IXE	RZE	TIL	Total
**SOC—General disorders and administration site conditions**	2	24	25	6	6	6		1	2		1	**72**
PT—Hyperhidrosis		1										1
PT1—Asthenia	1	11	15	4	4	4			1			40
PT2—Hot flush		2										2
PT3—Pallor		7										7
PT4—Administration site reactions	1	3	7	1	1	1			1		1	16
PT5—Peripheral edema			2	1								3
PT6—Pyrexia					1	1		1				3
**SOC—Vascular disorders**		1	1									**2**
PT—Cyanosis			1									1
PT1—Hypotension		1										1
**SOC—Skin and subcutaneous tissue disorders**	2	2	3			1			2			**10**
PT—Folliculitis	1											1
PT1—Rash	1								2			3
PT2—Pruritus		2	1									3
PT3—Alopecia			1			1						2
PT4—Tingling feet/hands			1									1
**SOC—Ear and labyrinth disorders**		1	2		1	1						**5**
PT—Vertigo		1	2		1	1						5
**SOC—Nervous system disorders**		2	3	2	2	1	2			1		**13**
PT—Headache		2	3	2	2	1	2			1		13
**SOC—Infections and infestations**	1	2	3		1							**7**
PT—Candidiasis infection			2		1							3
PT1—Herpes virus infection	1	2	1									4
**SOC—Respiratory, thoracic, and mediastinal disorders**	1	5	6		2							**14**
PT—Pneumonia	1 *	0	2									3
PT1—Interstitial pneumonia		1	1									2
PT2—Nasopharyngitis		1	2		1							4
PT3—Benign respiratory tract neoplasm		1 *										1
PT4—Dyspnoea		2	1		1							4
**SOC—Investigations**		9	5	1	3							**18**
PT—Transaminases increased		6	1		1							8
PT1—Hepatitis C antibody positive		1										1
PT2—Carcinoembryonic antigen increased		2	1		1							4
PT3—Blood count abnormal				1								1
PT4—Haemoglobin decreased			2		1							3
PT5—Red blood cell sedimentation rate abnormal			1									1
**SOC—Blood and lymphatic system disorders**		4	1		2							**7**
PT—Lymphocytosis					1							1
PT1—Thrombocytopenia		3										3
PT2—Anaemia		1			1							2
PT3—Splenomegaly			1 *									1
**SOC—Gastrointestinal disorders**		2	14	1	1	3			1			**22**
PT—Vomiting			2									2
PT1—Gingivitis		1		1								2
PT2—Nausea		1	5		1							7
PT3—Diarrhoea infectious			7			3			1			11
**SOC—Immune system disorders**	5	1										**6**
PT—Allergic reaction to excipient	5											5
PT1—Lupus-like syndrome		1 *										1
**SOC—Renal and urinary disorders**		1			1							**2**
PT—Haematuria					1							1
PT1—Haemorrhagic Cystitis		1 *										1
**SOC—Musculoskeletal and connective tissue disorders**			6	1	1							**8**
PT—Limb discomfort			1	1								2
PT1—Back pain			5		1							6
Total AEs	22	108	137	22	40	24	4	2	10	2	2	

* indicates the number of Serious Adverse events observed. Bold type is to differentiate the class of adverse events and distinguish it from subclasses. Abbreviations: IFX, infliximab; ETN, etanercept; ADA, adalimumab; GOL, golimumab; UST, ustekinumab; SEC, secukinumab; GUS, guselkumab; BRO, brodalumab; IXE, ixekizumab; RZE, Risankizumab; TIL, tildrakizumab; SOC, system organ class.

## Data Availability

The original contributions presented in the study are included in the article, further inquiries can be directed to the corresponding author/s.

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
