# Peer review of "Prescribing Pattern and Safety Profile of Biological Agents for Psoriasis in Real-World Practice: A Four-Year Calabrian Pharmacovigilance Analysis"

_pharmaceutics, 2024, doi:10.3390/pharmaceutics16101329_

Round 1
Reviewer 1 Report (Previous Reviewer 1)
Comments and Suggestions for Authors
The authors made the required corrections. It can be accepted now.
Author Response
We thank the Reviewer for the comments.
Reviewer 2 Report (Previous Reviewer 2)
Comments and Suggestions for Authors
The authors have, in my eyes, successfully adapted their manuscript. It has really improved compared to the first version which I evaluated as a reviewer. I recommend to accept it, after taking into consideration the following (minor) suggestions.
1. In the abstract both 'biologics' and 'biologicals' are used for the same. I advise aligning this, and also check the rest of the manuscript.
2. Line 181: mean - add 'age'
3. Tabel 1: explain Ps and PsA abbraviations in a footnote.
4. Table 1: comorbilities should be comorbitities.
5. Table 3: the legend can be clearer. The word 'per' is maybe less appropiate.
6. List of acronyms: the acronyms bDMARDs, csDMARDs and SAEs are plural, while the full meaning given is singular.
Author Response
Dear Reviewer,
We greatly appreciate your time and dedication during the evaluation of the manuscript and thank you for your comments. We have set out below our responses to the changes you requested. Our responses are listed in sequence and are highlighted in red in the revised version of this manuscript.
Comments to Reviewer 2
- We have aligned the manuscript by replacing biologics with biologicals.
- We have added ‘’age‘’ to line 181 as suggested (now line 172).
- We have explained the abbreviations Ps and PsA in a footnote in Table 1.
- We have replaced ‘’comorbilities’’ in ‘’comorbidities’’as suggested.
- We have changed the word 'per' in Table 3.
- We have corrected the list of acronyms.

Reviewer 3 Report (Previous Reviewer 4)
Comments and Suggestions for Authors
The manuscript is greatly improved. I congratulate the authors on an excellent resubmission.
Author Response
We thank the reviewer for the comments.
This manuscript is a resubmission of an earlier submission. The following is a list of the peer review reports and author responses from that submission.
Round 1
Reviewer 1 Report
Comments and Suggestions for Authors
1. Concise the first 2 statements in the abstract and focus more on the result part.
2. Many full forms missing for the abbreviated terms in Table 2.
3. What were the patient selection criteria? Any guideline followed.?
4. Any rationale for male/female criteria.
5. Can the obtained data be extrapolated with a large sample size?
6. What were the concerns during the follow-up period?
7. Revise the conclusion to include the key findings.
8. Discuss the results with suitable references (cross talk).
9. Check the numbering and formatting in Tables.
10. Briefly explain, in conclusion, the future scope of the current findings.
Reviewer 2 Report
Comments and Suggestions for Authors
The topic of the study by Caterina De Sarro is interesting but in the present form unacceptable for publication. The manuscript suffers from a poor structure, making the text often difficult to comprehend. Furthermore, the authors need to seek for professional English editing (which is offered also by MDPI). The text contains many syntax errors and therefore unclear sentences.
Title
· The current title suggests a very broad study (‘pharmacoepidemiological profile of biological agents for psoriasis in real-world practice’) while the span of the study is rather limited. Please think of another, more appropriate title.
Abstract
· The abstract is confusing and hard to understand. For instance, there seems a contradiction between lines 18-20 and 21.
· Structure the abstract along the line: introduction with clear aim (which is not visible now), methods used, main results and conclusions.
· The focus should be presented more clearly and the conclusion should connect to it (also counts for the manuscript as a hole).
Introduction
· There is quite some repetition in this part. Combine, for instance, information in line 36 with that in 42. The content is much alike.
· There is repetition in line 43 and 56 (QoL). Combine.
· The different forms of psoriasis presented is confusing. Align this in a better way.
· Line 60: international guideline, for what?
· Line 61: cs = conventional synthetic and not only synthetic.
· Line 74: used for which indication?
· Line 78, 79 2x ‘some’ which is not clear.
· Line 82: following treatment with biological agents (in general: be clearer in what you tell).
Materials and methods
· Line 85-86: five dermatology tertiary units? Please explain.
· Line 88-90 is incomprehensible.
· Line 92: GCP are not recommendations but guidelines.
· Section 2.1 lacks structure. Use paragraphs and discuss one topic per paragraph.
· Lines 120 ff is discussion and be transferred to that part of the manuscript.
· The last part of 2.1 is difficult to follow.
· Line 144-145 is cryptic.
· Line 146: what are ‘constant records’?
Results
· Replace decimal commas by points (in the text and in the tables).
· In the Results section there is a lot of repetition of figures that are also shown in tables. Avoid such repetition.
· 3.3, line 192: I cannot place ‘training time’.
· Legend to table 3: be consequent in using terminology. Not biological and non-biological agents but scDMARDs and bDMARDS as used elsewhere.
· Table 4: ADA/ETN (3)(2)??
Discussion and conclusion
· In a good discussion data from the study are discussed in relation to existing literature. The authors now rather review the literature. What do your findings mean in a broader context. The discussion should be entirely rewritten.
· Also, create a better and clearer structure by using more paragraphs.
· Do not repeat results here as done in lines 333ff.
· I would start the discussion with a short general introductory sentence, such as now given in lines 338-339.
· Line 339: what precisely is ‘long-term’?
· Write a clear conclusion. It is now a bit hidden. You may use a separate heading. Let the conclusion connect to the aim of the study in the introduction (which is not very now).
Comments on the Quality of English Language
The manuscript needs extensive rewriting.
Reviewer 3 Report
Comments and Suggestions for Authors
Reviewer Comments to Author(s)
Recommendation: Major revisions
This article provides information on the clinical application of biological agents in psoriasis patients, including TNF-α and IL-6 antagonists, IL-17A, IL-17, IL-12/23, and IL-23 inhibitors, and T-cell co-stimulation inhibitor. The article provides evidence of adverse effects (AEs) and serious AEs. The authors may consider the following.
1. In line 39 the authors should either include the word “including” within the parentheses or emit the use of them ().
2. It is important the phrases used to be easily readable and understandable. Please check and rephrase if needed the syntactic of the sentences in lines 44-46 “Plaque … response” and lines 52-55 “Therefore, … patients”. Please avoid long and extended sentences so to provide a clear meaning and use “,” when appropriate. For example, after “associated with general patient …” in line 53 the meaning and grammar are confusing. Moreover, what do the authors mean by “heterogeneous manifestation” and “general patients” since general patient is normally associated with health history and general health assessment.
3. In line 85 please check on the grammar may it should be “were treated with”. Moreover, the inclusion criteria are those referred to the next sentence lines 86-88 “Demographic … were stored”? It would be beneficial to specify.
4. What CBPP stands for? Please provide the description of the abbreviation the first time present in the text.
5. In line 111 that the “index date” is being described, the authors specify that is the date of initiation of the biological therapy within their protocol independent of naïve or with a previous biological agent. There is a question for the authors. Does and how the presence of patients that had already started a biological agent and initiated a new one affect their results? Are these patients prompt to increased or decreased AE or it does not affect at all the therapeutic outcome? The question refers to the patients that were following a biological therapy before the study protocol of the authors and not the patients that switched therapy within the period of the study. According to Table 1 from the 733 total number of patients the 353 were N. Patients (N for naïve?) at the index date. So the rest 380 were patients that before taking part in this study they were medicated another biological therapy or another therapy. Is this assumption correct and how is it expected to affect the tendency or not to therapy and AE or serious AEs? In line 180 it is described that 571 patients were naïve to biological therapy. Can the authors please clarify the N. Patients in the Table and the number of naïve patients at day 1 of their protocol?
6. Table 3 is not described at all, since the authors from Table 2 continue in describing Table 4. Please provide Results description of Table 3.
7. Please check the clarity and syntax of lines 293-295 of the sentence “Various unprompted reporting treatments carried out systems during phase IV analysis might offer interesting data on tolerability … appearance”
8.Line 305 please use the same abbreviation for PsA and in lines 338, 342.
9. Line 312 please correct “by surveys. anomalies.”
10. TNFs play a major part in the regulation of immune cells. By interacting with TNFR1 and TNFR2 are able to regulate pro-inflammatory and anti-inflammatory signaling pathways, respectively. TNF-α inhibitors are mainly used the past years in autoimmune disorders, including psoriasis. Definitely, the site of action of TNF-α inhibitors may affect either the pro-inflammatory or the anti-inflammatory signaling cascade. Probably that is the reason that in this study the TNF-α inhibitors provided upper respiratory tract AEs and affect the rate of infections among patients. Do the authors think that by providing same more explanations on the reason behind the observed AEs/SAEs would highlight the results of their study? This comment is advised since the Discussion part lacks of explanations on how the AEs or SAEs are linked to the type of biological therapy, since it is stated from the beginning that there are specific antagonists and inhibitors examined.
Comments on the Quality of English LanguageThe authors use long sentences and they should revise the clarity in such sentences. Details are provided in the comments.
Reviewer 4 Report
Comments and Suggestions for Authors
De Sarro et al. present a prospective investigation of the clinical administration patterns and outcomes in patients treated with biological agents from 2017-2021 in multiple centers in Italy.
The authors need to clarify if this was a prospective, observational investigation or a retrospective investigation of a data base generated from the information from all the centers identified. There was an institutional approval obtained, and the text of the article stated that patients were enrolled. Did they provide written, informed consent? Or were these data retrospectively obtained?
The rationale for the investigation is very valid – the work sought to take a “snap shot” of therapy, patterns of changes in therapy based on adverse events or ineffectiveness of therapy, and overall effectiveness of the multiple biologicals that have been brought to clinical practice. In short, the paper helps tell us where we are.
The methods used to include/exclude patients and analyze drug administration and outcome are standard and validated.
The data presentations were very informative and granular in detail. Table 4, in particular, provided information concerning the switching of therapies in a clear manner that showed patterns that would be of interest to clinicians.
The conclusions drawn are reasonable, based on the data.
The major comments this reviewer has is as follows:
1. Clear identification as to what type of investigation this is – prospective, observational or retrospective.
2. Volume of acronyms. The authors should strongly consider a table of acronyms as it would improve the ability of the reader to more easily understand the message of the authors as they consider the data.